# Intra-individual variation of particles in exhaled air and of the contents of Surfactant protein A and albumin

**Spela Kokelj**[1]*, **Jeong-Lim Kim**[1], **Marianne Andersson**[1], **Gunilla Runström Eden**[1], **Björn Bake**[2], **Anna-Carin Olin**[1]

1 Occupational and Environmental Medicine, School of Public Health and Community Medicine, Institute of Medicine, Sahlgrenska Academy, University of Gothenburg, Gothenburg, Sweden, 2 Unit of Respiratory Medicine and Allergy, Department of Internal Medicine, Institute of Medicine, Sahlgrenska Academy, University of Gothenburg, Gothenburg, Sweden

* spela.kokelj@amm.gu.se

## Abstract

### Introduction

Particles in exhaled air (PEx) provide samples of respiratory tract lining fluid from small airways containing, for example, Surfactant protein A (SP-A) and albumin, potential biomarkers of small airway disease. We hypothesized that there are differences between morning, noon, and afternoon measurements and that the variability of repeated measurements is larger between days than within days.

### Methods

PEx was obtained in sixteen healthy non-smoking adults on 11 occasions, within one day and between days. SP-A and albumin were quantified by ELISA. The coefficient of repeatability (CR), intraclass correlation coefficient (ICC), and coefficient of variation (CV) were used to assess the variation of repeated measurements.

### Results

SP-A and albumin increased significantly from morning towards the noon and afternoon by 13% and 25% on average, respectively, whereas PEx number concentration and particle mean mass did not differ significantly between the morning, noon and afternoon. Between-day CRs were not larger than within-day CRs.

### Conclusions

Time of the day influences the contents of SP-A and albumin in exhaled particles. The variation of repeated measurements was rather high but was not influenced by the time intervals between measurements.

**Data Availability Statement:** All relevant data are within the manuscript and its Supporting Information files.

**Funding:** This research was supported by the Swedish Heart-Lung Foundation (ACO, grant number 20180209). The funder had no role in the study design, data collection and analysis, decision to publish, or preparation of the manuscript.

**Competing interests:** I have read the journal's policy and the authors of this manuscript have the following competing interests: Björn Bake, Marianne Andersson and Gunilla Runström Eden are shareholders of PEXA® AB (www.PEXA.se) and Anna-Carin Olin is a board member (patent WO2009045163). This does not alter our adherence to PLOS ONE policies on sharing data and materials.

## Introduction

Particles in exhaled air (PEx) offer a new opportunity to monitor small airways, i.e., airways with an internal diameter $\leq$2 mm. Airway closure and subsequent opening produce particles (i.e., droplets) [1, 2]. The amount of airway closure is the greatest when an individual exhales to residual volume and then holds his/her breath for several seconds [1], and the particles generated during the subsequent inspiration contain a sample of respiratory tract lining fluid (RTLF) from the region where the airways close. During exhalation back to functional residual capacity, these particles are exhaled and those in the size range of 0.5–4 μm can be counted and sampled. Where airway closure and opening occur is still not fully elucidated, though in healthy individuals they are assumed to occur in the transition zone between the conductive and acinar airways, based on experimental data from dogs [3].

RTLF, an important protective interface between the respiratory epithelium and the external environment, consists mainly of surfactant [4, 5], comprising, among others, a major surfactant protein Surfactant Protein A (SP-A) and albumin, both of which are potential biomarkers of small airway disease [5, 6]. The number of particles exhaled is another potential biomarker of small airway inflammation, as it has been shown to be lower in subjects with chronic obstructive pulmonary disease (COPD) and asthma than in healthy subjects [4, 7].

Despite use of a standardized breathing maneuver, the number of exhaled particles varies considerably between individuals [6, 8]. Although age as well as anthropometric and spirometric variables explain 28–29% of inter-individual variation, the remaining variation is still very large [6]. As these measurements took place at various times during the day [6], it is possible that diurnal variation contributes to some extent to the variability of exhaled particles. The possibility of exhaled particles and their protein content being subjected to diurnal variation has not been investigated previously. Schwarz et al. [8] reported that the intra-individual variation was much smaller than the inter-individual variation of exhaled particles. The intra-individual variability for all lung function measurements tends to increase with increasing time interval between the measurements [9]. The intra-individual variation of exhaled particles, SP-A and albumin within one day and between days has not been studied before.

The aim of this study was to test the hypotheses that exhaled particles, SP-A and albumin are subjected to diurnal variation and that the variation between days is larger than the variation within one day. With these analyses, the repeatability of PEx variables will be described, allowing for comparison with other methods.

## Methods

Sixteen healthy volunteers, eight men and eight women aged 22–69 years, were recruited for the study using posted notices. Eligible for inclusion in the study were never- or ex-smoking individuals with no reported hay fever and no history of respiratory disease such as asthma and COPD, with those who had not smoked in the past six months being defined as ex-smokers. Exclusion criteria were current ongoing upper respiratory tract infection and pregnancy. Participants provided written informed consent before the measurements and the Regional Ethics Committee at the University of Gothenburg (030–18) approved the study. Table 1 presents the characteristics of the 16 participants.

### Study design

Each individual was examined on eleven occasions on five non-consecutive days. On three days, measurements were made in the morning (08:00–10:30), at noon (11:00–14:00), and in the afternoon (14:30–17:00). The exception was one individual for whom measurements were made at noon and in the early and late afternoon on two of the three days. On the other two

**Table 1. General characteristics and spirometry results of the participants.** Mean values and ranges are presented.

| | Women (n = 8) | | | Men (n = 8) | | |
|---|---|---|---|---|---|---|
| | **Mean** | **Min** | **Max** | **Mean** | **Min** | **Max** |
| Age (yrs) | 46.4 | 22 | 68 | 44,8 | 24 | 68 |
| Height (cm) | 165.9 | 157 | 177 | 180.8 | 173 | 191 |
| BMI (kg/m$^2$) | 22.1 | 17.9 | 26.2 | 24.5 | 21.9 | 28 |
| FVC (% pred) | 97.6 | 83.2 | 105.5 | 98.3 | 91.9 | 110.4 |
| FEV$_1$ (% pred) | 95.5 | 85.3 | 106.3 | 94.2 | 82.2 | 98.6 |
| FEV$_1$/FVC (%) | 78.3 | 71.1 | 86.8 | 76.6 | 69.2 | 85.6 |

days, only one measurement was made per day, any time between 08:00 and 17:00. The order of the days was random and the overall measurement period varied between five and 24 weeks. Table 2 presents the individual time intervals between the five days. The measurements were made between March and October 2018.

## Spirometry

Spirometry was performed using a Spirare spirometer (Spirare, Stockholm, Sweden) in accordance with the ATS/ERS criteria [10]. Forced vital capacity (FVC) and forced expired volume in one second (FEV$_1$) were expressed as a percentage of the reference value (% pred) according to Brisman et al. (note the corrigendum) [11].

## Exhaled particles

The equipment for counting exhaled particles, the PExA 2.0 instrument (see S2 Fig) (PExA AB, Gothenburg, Sweden), has been described previously in detail [12], though minor

**Table 2. Individual time intervals from the first investigation.**

| | | Number of days from first investigation | | | |
|---|---|---|---|---|---|
| id | Date of first investigation | Second invest. | Third invest. | Forth invest. | Fifth invest. |
| 1 | **2018-03-14** | **33** | **49** | 54 | 61 |
| 2 | **2018-03-19** | **31** | 120 | 157 | **162** |
| 3 | **2018-03-20** | **22** | **43** | 63 | 84 |
| 4 | **2018-03-20** | **22** | **35** | 44 | 58 |
| 5 | **2018-04-10** | **14** | **36** | 41 | 48 |
| 6 | 2018-04-18 | **15** | **33** | **55** | 63 |
| 7 | 2018-05-07 | **15** | 32 | **36** | **43** |
| 8 | 2018-05-07 | **7** | 14 | **21** | **35** |
| 9 | **2018-05-15** | **15** | **27** | 37 | 43 |
| 10 | 2018-06-08 | **6** | **11** | **18** | 39 |
| 11 | 2018-07-12 | **6** | **11** | 63 | **106** |
| 12 | **2018-08-21** | **15** | **21** | 30 | 35 |
| 13 | 2018-08-29 | **12** | 27 | 40 | **71** |
| 14 | 2018-09-04 | 6 | **21** | **28** | **50** |
| 15 | **2018-09-10** | **25** | 30 | 39 | **66** |
| 16 | 2018-09-13 | 12 | **20** | **28** | **35** |

**Bold characters** indicate the days when investigations were performed in the morning, noon and afternoon. On the other days the investigations were performed only once but at various times.

modifications have since been made [4]. The subject breathed via a mouthpiece and a two-way, non-re-breathing valve into the instrument, which consists of a thermostated box (36˚C) containing an optical particle counter (Grimm model 1.108; Grimm Aerosol Technik GmbH & Co., Ainring, Germany) and an impactor (Dekati Ltd., Tampere, Finland). The measured particles were between 0.41 and 4.55 μm in diameter and were sampled by impaction on a hydrophilic silicon wafer inside the impactor. The number of particles sampled on the silicon wafer was estimated based on measurements from an optical particle counter. Subjects inhaled HEPA-filtered air for a minimum of three breaths before sampling to remove the particles from ambient air. All subjects wore a nose clip throughout the procedure. A standardized breathing maneuver was used [1, 13], starting with an exhalation at the normal flow rate to residual volume, breath-holding for five seconds, followed by a maximal inhalation to total lung capacity, immediately followed by a normal exhalation to functional residual capacity. Exhalation flow was not controlled and was measured using an ultrasonic flow meter (OEM flow sensor; Spiroson-AS, Medical Technologies, Zürich, Switzerland), enabling visualization of the expiratory flow and volume [4]. Between breathing maneuvers, the subject breathed particle-free air tidally for 30–60 seconds. Each sampling session continued until 120 ng had been collected. The median number of breathing maneuvers required to collet 120 ng was 7, with a range from 4 to 53. The same investigator performed all sampling sessions. PEx number concentrations are expressed as $n \times 1000$ per liter of exhaled air (kn L$^{-1}$) and per breath (kn breath$^{-1}$). The particle mean mass (pg) estimated the particle size distribution [6].

## Chemical analysis

Before analysis, samples were extracted from PTFE membranes using 140 μL of extraction buffer with a composition of 10 mM PBS containing 1% BSA w/v and 0.05% TWEEN-20 v/v (Thermo Scientific, Rockford, IL, USA). Extraction was conducted for 60 min using a thermo-mixer (Comfort, Eppendorf AG, Hamburg, Germany) set at 37˚C and 400 rpm. The extracted sample volume was split: 40 μL each for SP-A and albumin, with the rest as a backup. The samples were quantified using an enzyme-linked immunosorbent assay (ELISA). The manufacturer's instructions for SP-A ELISA (BioVendor, Brno, Czech Republic) and albumin ELISA (E-80AL; Immunology Consultant Laboratories, Portland, OR, USA) were used with minor changes. Buffer (i.e., extraction buffer:assay diluent buffer, 1:2 ratio) was prepared for standards and controls to match the sample buffer composition. Before analysis, 80 μL of assay diluent buffer was added to the PEx samples. The first incubation time for SP-A was two hours at 37˚C with agitation at 300 rpm. Albumin was incubated for one hour at room temperature with agitation at 300 rpm. The reaction time was nine minutes. The precision of the assays was monitored by using three sample replicates in each run. The CVs of the replicates for SP-A and albumin were 0–10% with median CVs of 3% for SP-A and 4% for albumin and no significant difference between the three assays. SP-A and albumin results were expressed in weight percent (wt%), i.e., mass divided by the total particle mass and expressed as a percentage.

## Statistical analyses

According to the ATS/ERS task force, "Standardisation of lung function testing" [9], the coefficient of repeatability (CR) is recommended as the optimal method to assess intra-individual variability. Other widely used statistical tools are the intraclass correlation coefficient (ICC) [14] and coefficient of variation (CV) [9]. CR calculations were based on the differences between measurements, i.e., subtracting a later from an earlier measurement. The distribution of the differences was inspected visually using histograms and by comparing mean and median values. SP-A and albumin differences were normally distributed, while the PEx number

concentration differences were right skewed and therefore log transformed, using the natural logarithm (lnPEx), to follow a normal distribution. Particle mean mass differences were expressed as the percentage of the mean of the corresponding measurements (calculated by subtracting a later from an earlier measurement and dividing it by the mean of the two measurements) to achieve a normal distribution. CR was calculated by multiplying the standard deviation (SD) of the differences by 1.96. The two-way repeated ANOVA was tested for differences between the morning, noon, and afternoon values and between the three days with multiple measurements.

**Differences between morning, noon, and afternoon.** Only the measurements made three times in one day repeated on three different days were used for these calculations. For one individual lacking morning measurements on two days, only noon and late afternoon measurements were included in the analyses.

**Variation of repeated measurements within and between days.** Within-day CR calculations used differences between morning, noon, and afternoon measurements for each day for each individual, i.e., three differences per individual and day. Within-day ICC calculations were made for each of the three days separately and the average of the three days is presented. Within-day CV was calculated for each individual using three days with three measurements each and the average within-day CV and SD are presented.

To assess variation of repeated measurements between days, all five days were used and only noon measurements were used for days with multiple measurements. To calculate between-day CR, differences between all five measurements for each individual were calculated, i.e., a later measurement was subtracted from an earlier one, resulting in 10 differences per individual. Between-day ICC and CV were also calculated.

Microsoft Excel 2016 was used to calculate the CR and CV, and IBM SPSS Statistics for Windows, version 25 (IBM Corp., Armonk, NY, USA) was used to calculate two-way repeated ANOVA, ICC, and to create figures. The significance level was set to two-tailed p-value $< 0.05$.

## Results

### Differences between morning, noon, and afternoon

There were no significant differences between morning, noon, and afternoon regarding lnPEx (kn L$^{-1}$ and kn breath$^{-1}$) and particle mean mass (see Table 3). A small but significant within-

**Table 3. Results of the within day measurements.** Mean values and 95% confidence intervals are presented for the normally distributed lnPEx, particle mean mass expressed as a percentage of the mean of the inherent measurements, SPA and albumin. Median values and the interquartile range are given for the non-normally distributed PEx and particle mean mass.

|  | Mean (95% CI) or median (interquartile range) | | |
| --- | --- | --- | --- |
|  | **Morning** | **Noon** | **Afternoon** |
| PEx (kn/L)* | 12.8 (9.3–18.0) | 14.7 (9.6–18.1) | 14.7 (10.0–19.7) |
| lnPEx (kn/L) | 2.6 (2.3–2.8) | 2.6 (2.4–2.8) | 2.7 (2.4–3.0) |
| PEx (kn/breath)* | 40.7 (30.4–75.2) | 49.8 (28.8–65.3) | 54.9 (30.1–78.7) |
| lnPEx (kn/breath) | 3.8 (3.5–4.1) | 3.8 (3.6–4.1) | 3.9 (3.6–4.2) |
| Particle mean mass (pg)* | 0.46 (0.37–0.63) | 0.42 (0.33–0.52) | 0.42 (0.32–0.50) |
| Surfactant protein A (wt%) | 4.5 (3.8–5.2) | 5.1 (4.4–5.8) | 5.1 (4.3–5.9) |
| Albumin (wt%) | 5.4 (4.7–6.0) | 6.7 (5.8–7.6) | 6.7 (5.9–7.4) |

* indicates the variables that are presented as median values and interquartile range.

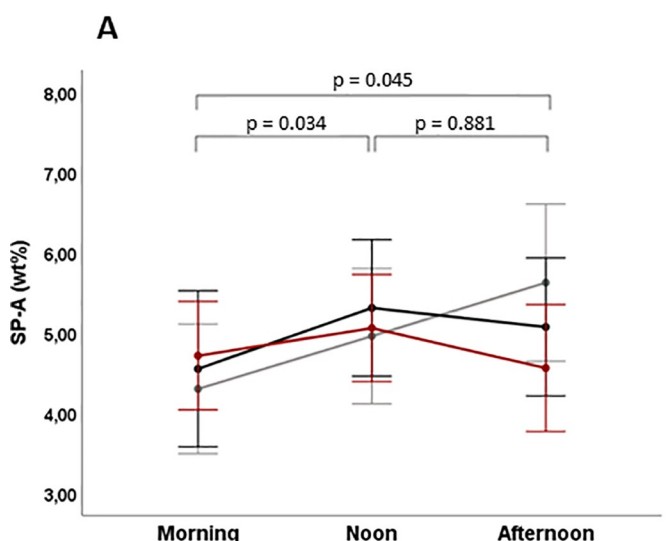
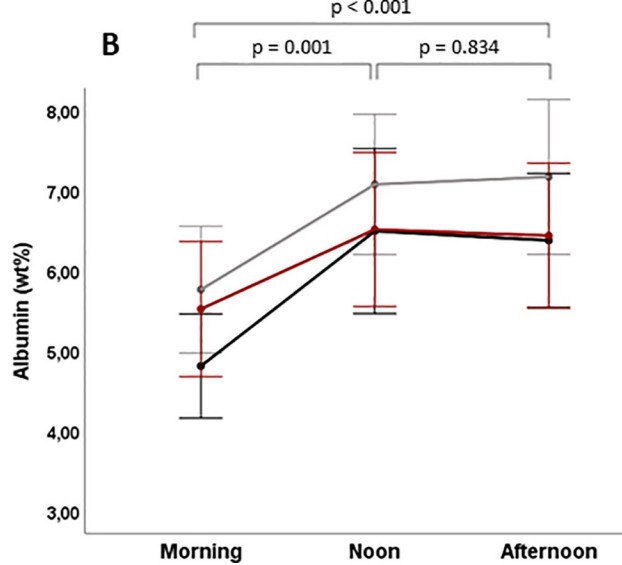

**Fig 1. Plots illustrating results from repeated measurements of SP-A and albumin within a day.** Each line represents one of the three days with multiple measurements (grey for day 1, black for day 2 and red for day 3). The dots represent mean values of the 16 subjects, and the error bars the corresponding 95% confidence intervals. P-values are according to two-way ANOVA for repeated measures.

day variation of SP-A was observed (p = 0.022; see Fig 1A). The average increase in SP-A from morning to noon was 13%, with the lowest values of SP-A in the morning (see Table 3 and Fig 1A). Albumin followed the same pattern as SP-A, with the lowest values in the morning and an increase towards noon (p = 0.001; Table 3 and Fig 1B). The average increase in albumin from morning to noon was 25%.

## Variation of repeated measurements within and between days

Table 4 presents within- and between-day CRs and means of the differences. Only the between-day mean of the differences for SP-A was not statistically significantly different from zero (p = 0.359; Table 4). Within-day ICC values were similar on the three days with multiple measurements; Table 5 presents the average of the three days as well as the between-day ICCs. Table 6 presents the average intra-individual CV for within- and between-day variation. Fig 2 illustrates the variation of repeated measurements between days; graphic representation reveals no important effect of the time interval of 60 days on the differences.

**Table 4. Coefficients of repeatability (CR) and means of the differences of the normally distributed variables lnPEx, particle mean mass percentage, surfactant protein A and albumin.**

|  | Within day | | Between days | |
| --- | --- | --- | --- | --- |
|  | **CR** | **Mean of the differences** | **CR** | **Mean of the differences** |
| lnPEx (kn/L) | 0.63 | -0.08* | 0.85 | -0.10* |
| lnPEx (kn/breath) | 0.72 | -0.08* | 0.96 | -0.11* |
| Particle mean mass (%) | 39 | 5* | 54 | -5* |
| SP-A (wt%) | 2.5 | -0.3* | 2.6 | -0.1 |
| Albumin (wt%) | 2.8 | -0.8* | 2.9 | 0.6* |

* indicates a statistically significant difference from zero.

**Table 5. Intraclass correlation coefficients (ICC).**

|  | Within day | Between days |
|---|---|---|
| PEx (kn/L) | 0.75 | 0.5 |
| PEx (kn/breath) | 0.75 | 0.68 |
| Particle mean mass (pg) | 0.73 | 0.69 |
| SP-A (wt%) | 0.64 | 0.66 |
| Albumin (wt%) | 0.55 | 0.59 |

## Discussion

The present study shows that SP-A and albumin levels are lower in the morning than at noon and in the afternoon, whereas the PEx number concentration and particle mean mass appear unaffected by the time of day. Furthermore, it shows that variation of repeated measurements of exhaled particles and their contents does not increase with increasing time interval between the measurements.

The PEx method samples RTLF from small airways while avoiding invasive techniques and unknown dilutions, as in, for example, bronchoalveolar lavage (BAL), induced sputum, and exhaled breath condensate [15–17]. However, the method has some drawbacks, namely the sample not containing any cellular material that can be obtained will BAL. RTLF from small airways contains phospholipids (approximately 90%) and proteins (approximately 10%). SP-A, synthesized almost exclusively by alveolar type-II cells, is the most abundant protein and is important for both host defense and surfactant function [18, 19]. Albumin is the major blood protein and is abundant in RTLF, constituting about 25% of the proteins in PEx samples [5], and may leak from plasma into the airways [20]. Surfactant protein D (SP-D) and Club (Clara) cell protein (CC-16) are present in high concentrations in RTLF and are considered markers of epithelial injury and COPD [21–24]. These potential biomarkers appear in serum as well, possibly due to passive leakage across the lung epithelium into the systemic circulation. However, SP-A, SP-D, and CC-16 are to a minor extent produced by extra-pulmonary tissues as well [25–27], but the correlations between lung and serum levels are weak [28–30].

The within-day variation of SP-A and albumin in exhaled particles agrees with biomarkers of lung injury in serum, such as SP-D and CC-16, which also display diurnal variation [26, 28]. The underlying mechanisms are not fully elucidated but likely concern diurnal differences in the production and secretion of these proteins by alveolar type-II cells and Club (Clara) cells or diurnal differences in transepithelial leakage due to changes in epithelial tight junctions [28]. Differences in leakage could potentially explain the diurnal variation of albumin in exhaled particles observed in the present study. Another potential reason for the observed diurnal variation in our study could be gravity, on account of the subjects' body position at night being horizontal compared to being vertical during the day, which may alter

**Table 6. Average of the intra-individual coefficients of variation (CV): Standard deviations are given within parenthesis.**

|  | Within day | Between days |
|---|---|---|
| PEx (kn/L) | 29% (14%) | 27% (11%) |
| PEx (kn/breath) | 33% (16%) | 30% (12%) |
| Particle mean mass (pg) | 15% (5%) | 15% (9%) |
| SP-A (wt%) | 18% (5%) | 17% (7%) |
| Albumin (wt%) | 19% (6%) | 16% (6%) |

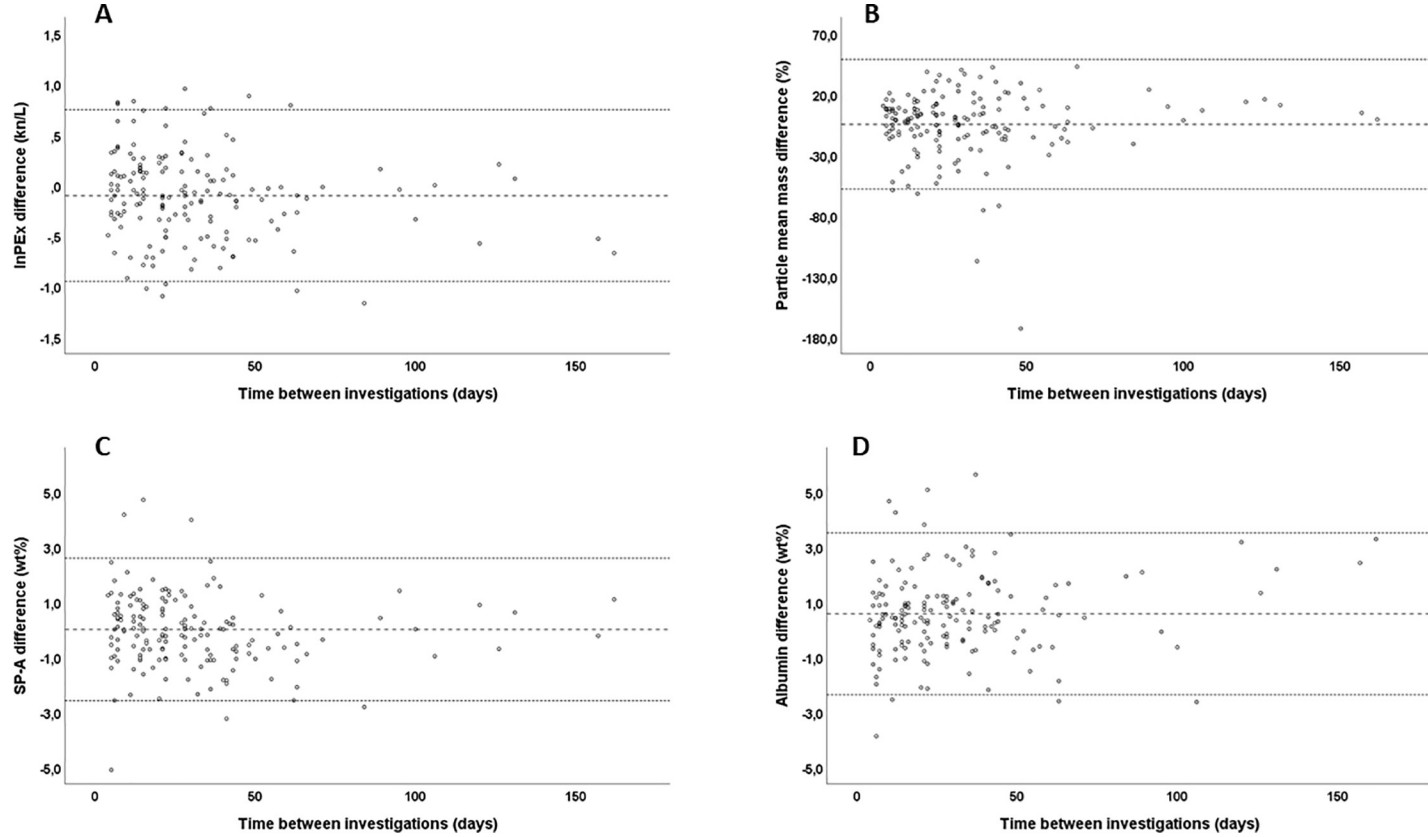

**Fig 2. The effect of the mean time interval between measurements on the corresponding differences between the five measurements of the 16 subjects.** The separate panels represent the differences of lnPEx (kn/L), of particle mean mass (the difference as a percentage of the mean of the corresponding measurements), of SP-A and of albumin. The lines represent the mean of the differences and the 95% upper and lower limits of agreement, as calculated by the mean of the differences ± CR.

mucocilliary transport in the airways. Furthermore, there is diurnal rhythm of spirometric variables in healthy subjects, with circadian minima at night and in the early morning hours [31, 32]. Borsboom et al. [32] reported a substantial increase in $FEV_1$, FVC, and peak expiratory flow from 09:00 to around noon, while total lung capacity hardly varied during the day, suggesting that substantial circadian changes may not occur in lung elastic recoil, but rather in airway caliber. On the other hand, the diffusing capacity of the lung and fractional exhaled nitric oxide do not seem to exhibit any diurnal variation [33–35]. The detailed mechanisms behind these differences are unclear. The circadian rhythm probably plays an important role in homeostasis and relates to the natural variations of wakefulness, body temperature, blood pressure, and hormone levels over a 24-h cycle and presumably to the variation of SP-A and albumin levels as well. The system consists of a central clock situated in the hypothalamus and peripheral clocks in each tissue and organ in the body [36].

A consequence of the lower levels of SP-A and albumin in the morning relative to noon and afternoon is that the time of day when taking a sample should be considered, both when comparing results within an individual and when comparing differences between groups.

The coefficient of repeatability is clinically useful: it is a precision index with the same units as the measurement tool and captures 95% of the differences between repeated measurements of the same subjects under stable conditions [14]. The between-day CR is clinically valuable, for example, when assessing an individual's change in lung function over time or following an

intervention, and the within-day CR is useful when interpreting provocation tests. Calculating the CR requires that the differences be normally distributed, and log or other transformations are sometimes required to achieve a distribution that is approximately normal. However, transforming the differences makes the clinical interpretation of the CR more challenging.

In the present study, the between-day CRs for the SP-A and albumin contents of exhaled particles were 2.6 wt% and 2.9 wt%, respectively. Thus, in a healthy subject with normal lung function, changes in SP-A and albumin levels between measurements made on different days should exceed 2.6 wt% and 2.9 wt%, respectively, before a clinically significant change should be considered. The difference between within- and between-day CRs was small and non-significant, indicating that the levels had been stable for at least two months. Our results are aligned with those of previous studies of long-term variation of SP-D and CC-16 in serum. Both SP-D and CC-16 in serum have been shown to remain stable over a three-month period in healthy individuals [25, 37]. Hoegh et al. found that serum SP-D levels remained stable even over a six-month period [26]. Lomas et al. reported a CR for serum CC-16 of 2.9 ng mL$^{-1}$ over three months [25]. The CRs for both SP-A and albumin in the present study are rather high, and the values are larger than the differences observed in cross-sectional studies between healthy subjects and patients with COPD and asthma [4, 7]. On the other hand, the calculation of CR in our study may overestimate the real variation in healthy subjects, since it was based on only 16 individuals and is therefore greatly influenced by the individuals with the highest variation.

ICC is a commonly used variability index that quantifies the strength of the relationship between measurements [14]. The main drawbacks of ICC are that it is difficult to interpret and is influenced by outliers [14, 38]. In a stable population, ICC values $\geq$0.6 are generally considered clinically useful [39]. Particle number concentration, particle mean mass, and SP-A levels were all reproducible with ICC values greater than 0.60. The within-day ICC for albumin, however, is low (0.55), while the ICC between the five days is higher, for both albumin (0.59) and SP-A (0.66). The explanation for this may be diurnal variation of SP-A and albumin levels resulting in higher within-day variability. In comparison, the ICC values of FEV$_1$ vary between 0.87 and 0.93 when measured by a trained investigator [34]. Exhaled nitric oxide measurements, used to monitor airway inflammation in asthma, also exhibit low intra-individual variability, with a within-day ICC above 0.94 [34]. Intra-individual variation of potential biomarkers measured in RTLF, however, has not been studied previously. Most studies have investigated the variation of biomarkers in serum or BAL. These studies on smokers and patients with COPD reported very small intra-individual variation of serum biomarkers (ICC for SP-D of 0.87 [21]), while the ICC values for biomarkers in BAL and sputum were lower (SP-D in BAL 0.63 [21], albumin in BAL 0.69, and albumin in sputum 0.61 [30]). This suggests that the reproducibility of the non-invasive sampling of biomarkers in PEx is comparable to that of other standard methods, such as BAL. This is surprising, since the PEx method samples RTLF directly from small airways, avoiding the unknown dilution problem that is present in BAL [15–17]. The reason for the relatively high variability of the SP-A and albumin contents in PEx is unclear, as the variability of the chemical analysis was presumably similar to the variability of the BAL fluid chemical analysis. The variability of the ELISA analysis was rather low in the present study for both proteins and can therefore explain only a minor part of the variability of SP-A and albumin in PEx.

Furthermore, it is surprising that the variability of SP-A and albumin contents is not substantially lower than the variability of the particle number concentration, since the proteins are expressed as the content percentage of the exhaled particles (wt%). However, there is very large variation of SP-A and albumin expressed in wt% at any given particle number concentration, revealing the lack of benefit of adjusting for PEx.

The present study also shows that variation does not increase when the time interval between measurements increases up to two months, in contrast to the repeatability of spirometry and fractional exhaled nitric oxide [9, 35, 40]. Furthermore, the high variability of PEx concentrations in certain individuals was not accompanied by changes in SP-A or albumin. The CV of particle number concentration (kn L$^{-1}$) between days was 27%, which is in good agreement with the results of Schwarz et al. [41], who reported an average intra-individual CV of 35% based on three measurements of exhaled particle number concentration.

One limitation of this study is its small number of subjects; on the other hand, we performed multiple measurements on each individual, which partly mitigates this. A second possible limitation is that the time intervals between the measurements differed between individuals.

In conclusion, SP-A and albumin in exhaled particles are subjected to diurnal variation, as opposed to particle number concentration and particle mean mass. The variation of repeated measures was uninfluenced by the increasing time intervals between measurements.

## Supporting information

**S1 Fig. Plots illustrating within-day intra-individual variation of particle number concentration, particle mean mass, SP-A and albumin for each individual (n = 16).** Each measurement was centered around each individual's mean ($yijk^* = yijk - \bar{yi}$, where i denotes an individual, j a day, k time of the day and $\bar{yi}$ the mean of all individual's i measurements) and the mean value of the three days for each individual is presented for each time point (morning, noon and afternoon). For SP-A (C) and albumin (D), p-values, calculated with two-way ANOVA for repeated measures for absolute levels of SP-A and albumin, are given.
(TIF)

**S2 Fig. Figure illustrating the PExA instrument.** Particle-free air is inhaled through a HEPA-filter. The subject breaths through a mouthpiece via a two-way, non-re-breathing valve into the instrument. A fraction of the air is characterized by an optical particle counter that operates at flow of 20 mL·s-1, while the remainder is drawn through the impactor. Particles are sampled from the aerosol with a two stage inertial impactor that is set up with a constant volumetric flow of 230 mL·s-1 (measured using an ultrasonic flow meter) using a RV pump (rotary vane pump). To handle exhalations exceeding the flow rate through the impactor, a reservoir that can buffer the exhaled air is used. A flow meter also measures the exhalation flow into the reservoir. All parts of the PExA instrument, except the mouthpiece, are in a thermostated box at 36˚C.
(TIF)

**S1 File. Data set.**
(PDF)

## Acknowledgments

The authors are grateful to all subjects for their participation in the present study. The authors would also like to thank Hatice Koca Akdeva for performing the chemical analysis and for the help with writing the Chemical analysis section of the manuscript.

## Author Contributions

**Conceptualization:** Jeong-Lim Kim, Marianne Andersson, Gunilla Runström Eden, Anna-Carin Olin.

**Formal analysis:** Spela Kokelj, Björn Bake.

**Investigation:** Marianne Andersson.

**Methodology:** Spela Kokelj, Björn Bake.

**Supervision:** Jeong-Lim Kim, Gunilla Runström Eden, Björn Bake, Anna-Carin Olin.

**Visualization:** Marianne Andersson.

**Writing – original draft:** Spela Kokelj.

**Writing – review & editing:** Spela Kokelj, Jeong-Lim Kim, Björn Bake, Anna-Carin Olin.

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
