## [Decision Letter · Decision Letter 0]

25 Nov 2019

PONE-D-19-30683

Intra-individual variation of particles in exhaled air and of the contents of Surfactant protein A and albumin

PLOS ONE

Dear Professor Spela Kokelj, 

Thank you for submitting your manuscript to PLOS ONE. After careful consideration, we feel that it has merit but does not fully meet PLOS ONE’s publication criteria as it currently stands. Therefore, we invite you to submit a revised version of the manuscript that addresses the points raised during the review process.

We would appreciate receiving your revised manuscript by Jan 09 2020 11:59PM. To enhance the reproducibility of your results, we recommend that if applicable you deposit your laboratory protocols in protocols.io, where a protocol can be assigned its own identifier (DOI) such that it can be cited independently in the future. For instructions see: http://journals.plos.org/plosone/s/submission-guidelines#loc-laboratory-protocols

We look forward to receiving your revised manuscript.

Kind regards,

Ajaya Bhattarai

Academic Editor

PLOS ONE

Journal Requirements:

1. Thank you for including your competing interests statement;  "I have read the journal's policy and the authors of this manuscript have the following competing interests: Björn Bake, Marianne Andersson and Gunilla Runström Eden are shareholders of PEXA® AB (www.PEXA.se) and Anna-Carin Olin is a board member. "

2. Thank you for including the following funding information within your acknowledgements section; "We wish to acknowledge assistance from the Centre for Allergy Research Highlights Asthma Markers of Phenotype (ChAMP) consortium, funded by the Swedish Foundation for Strategic Research, the Karolinska Institute, AstraZeneca & Science for Life Laboratory Joint Research Collaboration, and the Vårdal Foundation"

"This research was supported by the Swedish Heart-Lung Foundation (ACO, grant number 20180209). The funders had no role in study design, data collection and analysis, decision to publish, or preparation of the manuscript."

Additionally, because some of your funding information pertains to [commercial funding//patents], we ask you to provide an updated Competing Interests statement, declaring all sources of commercial funding.

In your Competing Interests statement, please confirm that your commercial funding does not alter your adherence to PLOS ONE Editorial policies and criteria by including the following statement: "This does not alter our adherence to PLOS ONE policies on sharing data and materials.” as detailed online in our guide for authors  http://journals.plos.org/plosone/s/competing-interests.  If this statement is not true and your adherence to PLOS policies on sharing data and materials is altered, please explain how.

Please include the updated Competing Interests Statement and Funding Statement in your cover letter. We will change the online submission form on your behalf.

Additional Editor Comments (if provided):

Major revision is necessary. After getting the revised manuscript, the further decision will be given.

Reviewers' comments:

Reviewer's Responses to Questions

**Comments to the Author**

1. Is the manuscript technically sound, and do the data support the conclusions?

Reviewer #1: Yes

Reviewer #2: Partly

2. Has the statistical analysis been performed appropriately and rigorously? 

Reviewer #1: No

Reviewer #2: Yes

3. Have the authors made all data underlying the findings in their manuscript fully available?

Reviewer #1: Yes

Reviewer #2: Yes

4. Is the manuscript presented in an intelligible fashion and written in standard English?

Reviewer #1: Yes

Reviewer #2: Yes

5. Review Comments to the Author

Reviewer #1: This paper reports findings from a small-sample size study. Given the novelty of the exhaled particle technique and the scarcity of the data on this topic, the results are interesting and important enough to be published despite that the results may be considered preliminary. The findings support the need for a more in-depth study with larger sample size to generate more generalizable results. Specific comments are as follows.

Abstract: CRs should be defined. As written, the units of CRs for different analytes are not normalized to mean values. e.g., reporting a CR as 0.85 kn L-1 gives the reader no idea on variability.

Table 1: Since this is not a randomly selected cohort of subjects, normal distributions would not be expected for any of the anthropocentric (and perhaps spirometric) parameters shown in Table 1. It is more informative to provide range, median, etc, in addition to mean and SD.

Methods: The method of exhaled particle collection should be described in more detail perhaps with a diagram of the sampling set up. How was the sampling air flow controlled? What sampling medium was used to collect particles? How long did it take to collect 120 ng particles? How was this monitored?

Statistical analyses section: the sentence starting "particle mean mass differences..." (Line 141-143) is unclear. Next sentence - what's the rationale to define CR as SD multiplied by 1.96? ANOVA tests presented in the paper are okay; but given the data structure, statistical testing for intra-day and between-day differences can be done more appropriately using mix-effects models, allowing control for subject as a random effect.

Table 3: the heading should be "mean(95% CI) or median (interquartile range)".

Discussion section is very well done.

Reviewer #2: This paper reports on measurements of exhaled breath from 16 subjects. It seems to be partly an advertisement for the PExA instrument (the authors state a conflict) and partly an evaluation of some of the contents of exhaled breath. The paper has some major flaws, discussed below, but the bottom line is that the hypothesis is not clearly stated. The hypothesis that I find is that intra-individual variation is small as measured by CR so that larger variations might indicate onset or resolution of disease. The paper should be restructured around this hypothesis or another one if my interpretation is incorrect.

In addition, the authors present the context of this work in terms of that of their prior co-authors. There is an extensive and growing literature on exhaled breath analysis. The introduction needs to be restructured to address the goals of this work in the context of other work in the field.

Major Comments

Line 47-48: this statement needs references.

Line 50-51: what is a PExA instrument? The introduction should describe the context for the work not the instrumentation used to perform the work. If the point of this work is to highlight the PExA instrument, it should be compared to other instruments that produce similar results.

Line 36: Define lnPEx. Is it the log of PEx?

Line 200-201: BAL contains information on cellular content of for instance neutrophils and eosinophils, which are important health markers. It would be very interesting if exhaled breath particles also contained cellular information, but this was not tested in this study. This sentence is more of a sales pitch for the PExA instrument and not a balanced evaluation of the pros and cons of exhaled breath analysis vs. BAL.

Lines 202-211: No analysis was performed of SP-D or CC-16 in this study. Why is it discussed here?

Line 218: this study did not include any subjects with asthma – why is this included here?

Line 223-228: speculation. It could be that gravity plays a role. At night the subjects were horizontal while in the day vertical, which may alter mucociliary transport and may explain some of the observations. This is also speculation.

Table 4: wt% might vary with relative humidity – drier conditions might lead to higher concentrations. Was RH in the clinic measured? If not, are these results valid?

6. PLOS authors have the option to publish the peer review history of their article (what does this mean?). If published, this will include your full peer review and any attached files.

Reviewer #1: Yes: Junfeng (Jim) Zhang

Reviewer #2: No

---

## [Author Response · Author response to Decision Letter 0]

10 Dec 2019

Dear Ajaya Bhattarai, 

Thank you for giving us the opportunity to submit a revised manuscript entitled “Intra-individual variation of particles in exhaled air and of the contents of Surfactant protein A and albumin”.

We appreciate the time and effort that PLOS ONE and the reviewers have dedicated providing valuable feedback on our manuscript and we have incorporated changes in the manuscript to reflect most of the suggestions provided by the reviewers. These changes are highlighted in a separate file. We consider this revision, incorporating reviewers’ insightful comments, to have substantially improved the manuscript.

Looking forward to hearing from you regarding our revision and to respond to any further questions and comments you may have that would improve the manuscript. 

Sincerely,

Spela Kokelj (on the behalf of the co-authors)

Journal Requirements:

1. Updated Competing Interests statement is provided in the cover letter and the patent number has been included.

2. The Funding Statement was incorrect and it has been revised. The funding related text has been removed from the manuscript and the Acknowledgements section has been changed (line 310-313).

Response to the Reviewer #1:

Comment: This paper reports findings from a small-sample size study. Given the novelty of the exhaled particle technique and the scarcity of the data on this topic, the results are interesting and important enough to be published despite that the results may be considered preliminary. The findings support the need for a more in-depth study with larger sample size to generate more generalizable results. Specific comments are as follows.

Reply: We appreciate the positive feedback from the reviewer. 

Comment: Abstract: CRs should be defined. As written, the units of CRs for different analytes are not normalized to mean values. e.g., reporting a CR as 0.85 kn L-1 gives the reader no idea on variability.

Reply: We agree with this comment. The abstract is rewritten to highlight diurnal variation and effect of time between the measurements on variation. CR is defined in the method section and CRs are instead presented only in Table 4 and Table 3 gives the reader an overview of mean values for all variables. 

Comment: Table 1: Since this is not a randomly selected cohort of subjects, normal distributions would not be expected for any of the anthropocentric (and perhaps spirometric) parameters shown in Table 1. It is more informative to provide range, median, etc, in addition to mean and SD.

Reply: We agree and have therefore revised Table 1 where we present the mean value and range for all variables. 

Comment: Methods: The method of exhaled particle collection should be described in more detail perhaps with a diagram of the sampling set up. How was the sampling air flow controlled? What sampling medium was used to collect particles? How long did it take to collect 120 ng particles? How was this monitored?

Reply: A figure of the PExA instrument and the sampling set-up has been added as a supplement (S2 Fig.) Exhalation flow was measured using an ultrasonic flow meter (OEM flow sensor; Spiroson-AS, Medical Technologies, Zürich, Switzerland), enabling visualization of the expiratory flow and volume. The sampling air flow was, however, not controlled, since the flow through the impactor is controlled and constant and the exhalations exceeding the flow rate are buffered in the reservoir. The sampling medium used to collect particles was a hydrophilic silicon wafer inside the impactor. We have included this in the Methods section under Exhaled particles (line 104). The median number of breathing maneuvers required to collect 120 ng was 7, with a range from 4 to 53. This has been added to the Methods section under Exhaled particles (line 114-115).

Comment: Statistical analyses section: the sentence starting "particle mean mass differences..." (Line 141-143) is unclear. 

Reply: Regarding the sentence about particle mean mass differences (line 141 -142) – the text has been revised. The sentence (line 143-146) now reads: “Particle mean mass differences were expressed as the percentage of the mean of corresponding measurements (calculated by subtracting a later from an earlier measurement and dividing it by the mean of the two measurements).” 

Comment: Next sentence - what's the rationale to define CR as SD multiplied by 1.96? 

Reply: The rationale behind CR = ±1.96*SD is that CR then defines the maximum difference which is likely to occur between measurements with 95% probability, assuming normal distribution.

Comment: ANOVA tests presented in the paper are okay; but given the data structure, statistical testing for intra-day and between-day differences can be done more appropriately using mix-effects models, allowing control for subject as a random effect.

Reply: As the reviewer mentioned above, we could apply mix-effects models allowing control for subjects as a random effect. Then, we can obtain the coefficients of predictor variables, which are adjusted for subjects as random. However, we are not only interested in coefficients of predictors but also of the effects of actual day- and daytime variance. Therefore, we treat them as fixed variables to yield the value of coefficients. 

Comment: Table 3: the heading should be "mean (95% CI) or median (interquartile range)".

Reply: The suggested correction in Table 3 has been made.

Comment: Discussion section is very well done.

Reply: The positive feedback is appreciated.

Response to the Reviewer #2:

Comment: This paper reports on measurements of exhaled breath from 16 subjects. It seems to be partly an advertisement for the PExA instrument (the authors state a conflict) and partly an evaluation of some of the contents of exhaled breath. The paper has some major flaws, discussed below, but the bottom line is that the hypothesis is not clearly stated. The hypothesis that I find is that intra-individual variation is small as measured by CR so that larger variations might indicate onset or resolution of disease. The paper should be restructured around this hypothesis or another one if my interpretation is incorrect.

In addition, the authors present the context of this work in terms of that of their prior co-authors. There is an extensive and growing literature on exhaled breath analysis. The introduction needs to be restructured to address the goals of this work in the context of other work in the field.

Reply: We thank the reviewer for the keen concern. However, we claim that the authors do not want to promote any commercial advertisement of a specific instrument and are only interested in evaluating the method of exhaled particles itself. Therefore, we have now revised the manuscript and the PExA® instrument is mentioned in the method section only, which is necessary in this context. We have also clearly stated a hypothesis in the introduction, as suggested (line 66-69).

Comment: Line 47-48: this statement needs references.

Reply: The reference has been added.

Comment: Line 50-51: what is a PExA instrument? The introduction should describe the context for the work not the instrumentation used to perform the work. If the point of this work is to highlight the PExA instrument, it should be compared to other instruments that produce similar results.

Reply: We have restructured the introduction to shift focus away from the PExA instrument and instead focus on the context of the work. We have also mentioned the work of the others (Schwarz et al.) that are using a similar method to sample exhaled particles. However, the results and the methods are hard to compare since there is nobody using this breathing maneuver that is used with our method. The results would be different if tidal breathing would be used like with other similar methods. On top of that, there is lack of methods that allow a similar chemical analysis of the exhaled particles to our method.

Comment: Line 36: Define lnPEx. Is it the log of PEx?

Reply: lnPEx is a natural logarithm of PEx. The abstract has been restructured and does therefore not include lnPEx. The definition of lnPEx has been added to the manuscript (line 143).

Comment: Line 200-201: BAL contains information on cellular content of for instance neutrophils and eosinophils, which are important health markers. It would be very interesting if exhaled breath particles also contained cellular information, but this was not tested in this study. This sentence is more of a sales pitch for the PExA instrument and not a balanced evaluation of the pros and cons of exhaled breath analysis vs. BAL.

Reply: You have raised an important point here, however, we do not agree with the comment on “a sales pitch”. The authors would like to emphasize that the aim of the present study was to evaluate the PEx method and not to promote the PExA instrument. To the best of our knowledge, there is no comparable method sampling exhaled particles. The exhaled particles obtained with our instrument unfortunately do not contain cellular information such as for example BAL. We have changed the sentence and it now reads (line 207-210): “The PEx method samples RTLF from small airways while avoiding invasive techniques and unknown dilutions, as in, for example, bronchoalveolar lavage (BAL), induced sputum, and exhaled breath condensate. However, the method has some drawbacks, namely the sample not containing any cellular material that can be obtained will BAL.”

Comment: Lines 202-211: No analysis was performed of SP-D or CC-16 in this study. Why is it discussed here?

Reply: They are other known components of respiratory tract lining fluid. We know nothing about variability of SP-A or albumin, since it has not been studied previously, we wanted to compare it to other known components of respiratory tract lining fluid that have been studied before, especially since they also exhibit diurnal variation and are secreted by the same cells as SP-A (for SP-D).

Comment: Line 218: this study did not include any subjects with asthma – why is this included here?

Reply: The suggestion has been noted and asthma has been omitted (line 228-229).

Comment: Line 223-228: speculation. It could be that gravity plays a role. At night the subjects were horizontal while in the day vertical, which may alter mucociliary transport and may explain some of the observations. This is also speculation.

Reply: Thank you for bringing this interesting point up. We agree with the comment and have incorporated it in the discussion section (line 225-228).

Comment: Table 4: wt% might vary with relative humidity – drier conditions might lead to higher concentrations. Was RH in the clinic measured? If not, are these results valid?

Reply: The humidity in the instrument is similar to the humidity of the air we are exhaling. The humidity in the instrument is kept constant throughout the measurement by the exhaled air that is saturated at 36 °C circling around. We assume the humidity of exhaled air is similar between subjects. Furthermore, the exhaled particles are not at any point in contact with room air, so they are not under the influence of the relative humidity in the room.

---

## [Decision Letter · Decision Letter 1]

6 Jan 2020

Intra-individual variation of particles in exhaled air and of the contents of Surfactant protein A and albumin

PONE-D-19-30683R1

Dear Dr. Kokelj,

We are pleased to inform you that your manuscript has been judged scientifically suitable for publication and will be formally accepted for publication once it complies with all outstanding technical requirements.

With kind regards,

Ajaya Bhattarai

Academic Editor

PLOS ONE

Additional Editor Comments (optional):

Reviewers' comments:

Reviewer's Responses to Questions

**Comments to the Author**

1. If the authors have adequately addressed your comments raised in a previous round of review and you feel that this manuscript is now acceptable for publication, you may indicate that here to bypass the “Comments to the Author” section, enter your conflict of interest statement in the “Confidential to Editor” section, and submit your "Accept" recommendation.

Reviewer #1: All comments have been addressed

2. Is the manuscript technically sound, and do the data support the conclusions?

Reviewer #1: (No Response)

3. Has the statistical analysis been performed appropriately and rigorously? 

Reviewer #1: (No Response)

4. Have the authors made all data underlying the findings in their manuscript fully available?

Reviewer #1: (No Response)

5. Is the manuscript presented in an intelligible fashion and written in standard English?

Reviewer #1: (No Response)

6. Review Comments to the Author

Reviewer #1: (No Response)

7. PLOS authors have the option to publish the peer review history of their article (what does this mean?). If published, this will include your full peer review and any attached files.

Reviewer #1: Yes: Junfeng (Jim) Zhang

---

## [Editor Report · Acceptance letter]

14 Jan 2020

PONE-D-19-30683R1 

Intra-individual variation of particles in exhaled air and of the contents of Surfactant protein A and albumin 

Dear Dr. Kokelj:

I am pleased to inform you that your manuscript has been deemed suitable for publication in PLOS ONE. Congratulations! Your manuscript is now with our production department. 

With kind regards,

on behalf of

Dr. Ajaya Bhattarai 

Academic Editor

PLOS ONE